# Migration of Dissolved Organic Matter in the Epikarst Fissured Soil of South China Karst

Kun Cheng [1,2] , Ziqi Liu [1,2], Kangning Xiong [1,2,*] , Qiufang He [3], Yuan Li [1,2], Lulu Cai [1,2] and Yi Chen [1,2]

1 School of Karst Science, Guizhou Normal University, Guiyang 550001, China; 20010170529@gznu.edu.cn (K.C.)
2 State Engineering Technology Institute for Karst Desertification Control, Guiyang 550001, China
3 Chongqing Key Laboratory of Karst Environment, School of Geographical Sciences, Southwest University, Chongqing 400700, China
* Correspondence: xiongkn@gznu.edu.cn

**Abstract:** The efficient reactivity and mobility of dissolved organic matter (DOM) affect biogeochemical processes. As important components that link aboveground and belowground vertical systems under the binary 3D structure of karst, fissures provide soil–water–nutrient leakage channels and storage spaces. However, reports on DOM properties and drivers in fissured soil are extremely rare. This study characterizes DOM in the fissured soil of different vegetation types under medium-intensity rocky desertification conditions. Soil samples were characterized via ultraviolet (UV)–visible absorption spectroscopy and fluorescence excitation–emission matrix–parallel factor analysis. Five fluorescent fractions were identified. The controlling factors for the optical properties of soil DOM were determined via the redundancy analysis method. Results showed the following: (1) Dissolved organic C/soil organic C < 4.68 + 0.49‰, specific UV absorbance $(SUVA)_{254}$ and $SUVA_{260}$ exhibited low overall performance with the vast majority of the humification index (HIX) < 4, most of the fluorescence index (FI) $\geq$ 1.7, most of the biological index (BIX) in 0.6 < BIX < 1 and 31.67–41.67% of protein-like fractions. These data indicate that cleaved soil, except for topsoil, has low DOM content, weak aromaticity, and low humification; (2) Rainfall intensity, aperture, and near-surface vegetation type are the major causes of DOM transport and loss; and (3) Most DOM losses are likely to be protein-like and enhance the loss of soil P. In summary, environmental factors and the characteristics of fissures determine DOM content and migration, particularly rainfall intensity and vegetation type. The loss of lighter DOM components will be greater in an area with high karst desertification grade, strong fissure development, weaker soil aromaticity, and lower humification. These results provide a clearer basis for optimizing the fissure nutrient element migration scheme in karst areas.

**Keywords:** DOM; UV; fluorescence; migration; karst; fissure

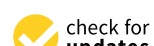



## 1. Introduction

Amongst the 17 sustainable development projects set by the United Nations to be achieved by 2030, 8 are related to the soil environment [1]. Soil plays a mediating and critical role in ecosystems and human society [2] with extremely low production rates [3]. DOM is the most dynamic and effective component of soil organic matter (SOM), and its strong reactivity and migration capacity play important roles in soil C cycling [4–6]. DOM can bind to soil particles to become part of the soil's organic carbon (SOC) pool [7]; it also promotes soil microbial activity, organic matter decomposition, nutrient transport and transformation in soil, weathering of minerals, soil-forming processes, and contaminant transport [8–13]. All these processes are related to DOM's properties and the characteristics of the environment.

DOM belongs to natural organic matter, accounting for about 97.1% [14]. It refers to C-based organic compounds [15] in soil and water bodies that consist of a range of molecules with different sizes and structures [16] that can pass through 0.1–0.7 μm filter membranes.

DOM is widely present in soil, water, the atmosphere, and sediments [17,18]. Studies have shown that DOM is a nonhomogeneous mixture of aliphatic and aromatic polymers whose composition varies in time and space depending on the source and the exposure to degradation processes [19–21]. In terrestrial environmental systems, most DOM in the soil is derived from plants, exogenous organic matter, and microbial components [9,22,23]. Studies have shown that the source of DOM determines its chemical properties and persistence in soil [5,24]. Thus, the characteristics of DOM and its sources in soil are particularly critical for determining DOM's environmental impact.

As an important part of the Earth's critical zone, the karst critical zone accounts for 12% of the total land area [25]. Amongst the karst critical zones, the epikarst is the product of intense karstification of the surface in the karst area, it is the first karst development layer below the surface, a transitional zone connecting the karst dual hydrological structure, the main space for the storage and migration of surface soil and water, and an important factor for controlling the development of karst geomorphology and eco-hydrological function [26]. In the context of karst desertification, the special binary three-dimensional spatial structure of karst areas [27] forms a special erosion mode as subsurface leakage. Subsurface leakage is the migration of surface soil to subsurface space through karst channels, such as dissolution fissures and waterfall holes, under the action of hydraulic transport and chemical dissolution in karst areas [28]. This phenomenon is the primary means of soil erosion under karst desertification conditions [29–31], making fissures an integral part of the vertical system that links aboveground and belowground in key karst zones, becoming one of the soil–water–nutrient loss channels and storage spaces. Water transport and nutrient transport in fractures are of considerable interest. Studies have concluded that the size of soil and water leakage along the pore (fissure) space depends on the development of the pore (fissure) space and the connectivity of the lower part of the fissure [31]. The connectivity of shallow karst fissures and soil differences exert a strong influence on the permeability of the surface karst zone [32,33], and the presence of preferential flow in fissured soil enhances nutrient loss [34]. Researchers [35] have conducted experiments under different subsurface fissure degrees and rainfall intensities by simulating artificial rainfall, and they have concluded that overall nutrient loss from sloping farmlands, such as N, P, and K, is not significantly related to fractality; moreover, rainfall intensity is a key factor, indicating that subsurface runoff is the primary nutrient loss mode on karst slopes [36]. DOM can be used as a tracer for understanding the hydrological pathways of soil nutrient transport [37]; it fully participates in the C cycle and enhances soil nutrient transport to the surrounding surface and groundwater environment [9,38,39]. In addition, subsurface migration and loss of DOM pose a greater threat to environmental quality and can considerably affect the transport and toxicity of organic and inorganic contaminants [40,41]. As reported in Sun [42], high-molecular-weight polycyclic aromatic hydrocarbons (PAHs) are used as carriers in soil seepage with DOM in the southwestern karst region, and PAHs continue to dissolve into seepage water when water is transported vertically downward along the profile, increasing the risk of groundwater contamination. By contrast, the surface of karst areas is fragmented and heterogeneous, and the distribution and migration factors of DOM in fractured soil are more variable. Therefore, the spectral information of chromophoric dissolved organic matter (CDOM) in DOM is used to reveal the characteristics and migration factors of DOM in karst fractured soils and provide a theoretical basis for improving the ecological environment of karst desertification.

We studied the content and spectral characteristics of DOM in typical fractured soils in the desertification areas in South China Karst, obtained the spectral factors of DOM by using ultraviolet–visible (UV–Vis) absorption and fluorescence spectroscopy, compared optical indices, performed parallel factor analysis (PARAFAC) and three-dimensional fluorescence excitation–emission matrices (3D-EEMs) to analyze the source and characteristics [43–47] of DOM and explore its migration control factors. The objectives of this study were to: (1) determine the distribution characteristics of DOM in fissured soils in karst areas; (2)

identify the drivers of DOM in fissured soil; and (3) assess the environmental impact of DOM transport in fissured soil.

## 2. Materials and Methods

### 2.1. Characteristics of the Study Area

In the mountainous plateau of Guizhou, which represents the general structure of the karst ecosystem in southern China, the Zhenfeng–Huajiang karst desertification demonstration area was selected as the study area (105°36′30″–105°46′30″ E, 25°39′13″–25°41′00″ N) [26,48]. The terrain is undulating, with an altitude of 600–1400 m. The study area (Figure 1) is a subtropical dry–hot valley climate, with a mean annual precipitation of 1052 mm and a mean annual temperature of 18.4 °C. Lithology is mostly limestone, with serious soil fragmentation, high rock exposure rate, and serious soil erosion, belonging to medium-intensity karst desertification [49]. Vegetation is largely broad-leaved forests, mixed coniferous forests, and shrubs. The primary vegetation has been severely damaged, and secondary vegetation is dominant at present. For details of the study area, refer to [34].

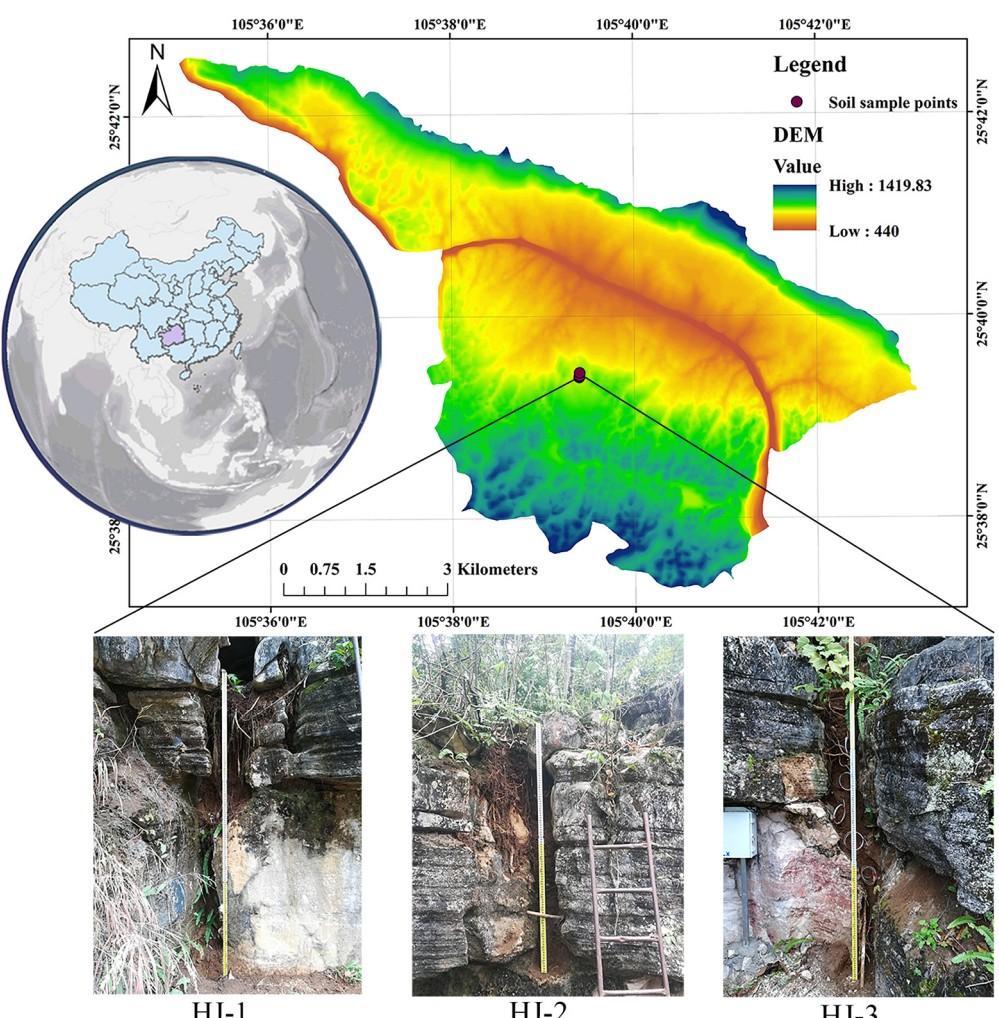

**Figure 1.** Locations of the study area and fissure soil sampling sites.

### 2.2. Sample Collection and Processing

The number of fissures, aperture, width, depth, dip angle, and near-surface vegetation were recorded through a survey of fissures in the study area. The characteristics of fissure development were clarified under varying topography and environment, and 3 typical fissures, proximity, and similar development were selected from amongst 65 fissures

(Table 1). Soil samples were collected in late August 2020 to remove loose soil, roots, and branches from the fissured soil. The soil was planed away from the outer 15 cm of the vertical profile and 1–2 cm away from the fissured rock wall. The ring knife was collected laterally from the in situ soil to measure its physical properties. Soil samples of organic matter to be measured are collected from 0–50 cm below the surface of the soil, once every 5 cm, and once every 10 cm after more than 50 cm. Sampling was performed at each layer, and 3 parallel soil samples were mixed for a total of 90 samples. The collected soil samples were stored in polyethylene (PE) self-sealing bags, transported back to the laboratory, cooled, and underwent ventilation air-drying.

**Table 1.** Overview of sampling sites and fissure structure characteristic parameters in the study area.

| Type | Longitude and Latitude | Altitude (m) | Depth (cm) | Aperture (cm) | Average Width (cm) | Dip Angle (°) | Near-Surface Vegetation |
|---|---|---|---|---|---|---|---|
| HJ-1 | 25°39′23.70″ N, 105°39′23.70″ E | 739 | 290 | 42 | 37 | 82 | Celtis sinensis and Cipadessa baccifera |
| HJ-2 | 25°39′24.56″ N, 105°39′23.82″ E | 739 | 190 | 31 | 25 | 87 | Eriobotrya japonica and Zanthoxylum bungeanum |
| HJ-3 | 25°39′26.42″ N, 105°39′23.96″ E | 739 | 285 | 39 | 32 | 80 | Celtis sinensis and Lonicera japonica |

For details of the physical properties (e.g., volume weight and total porosity) and mechanical composition of the soil, refer to [34]. The air-dried soil samples were ground and passed through a 60-mesh sieve. Then, 5 g of soil passed through the 60-mesh sieve was weighed, and 25 mL of ultra-pure water was added, shaken, and centrifuged to obtain the supernatant. The supernatant was filtered through a 0.22 μm polyethersulfone membrane (reducing the role of microbes on the DOM) to obtain the solution to be tested. Dissolved organic carbon (DOC) concentrations were determined using a total organic carbon (TOC) analyzer (multi N/C 3100, Analytik Jena, Jena, Germany). All dissolutions to be measured were uniformly diluted to a DOC concentration of less than 10 mg·L$^{-1}$ to ensure that absorbance at ultraviolet 254 nm was less than 0.3, reducing the internal filtering effect during fluorescence scanning [50]. The UV–Vis absorption spectra of the soil samples were obtained using a UV–Vis spectrophotometer (SPECORD Plus 200, Analytik Jena, Germany) with ultra-pure water as the blank set and scanning at 1 nm intervals from 200–800 nm. Fluorescence data from the soil samples were collected using a fluorescence spectrophotometer (RF-5301PC, Shimadzu, Kyoto, Japan). The excitation wavelengths (Ex) ranged from 220–500 nm at 5 nm intervals and the emission wavelengths (Em) ranged from 250–600 nm at 1 nm intervals, minus ultra-pure water as a control to eliminate scat-tering. Additional information is found in Lawaetz and Stedmon [51] and He [44].

### 2.3. Optical Factor

As shown in Table 2 for the DOM optical indicators used in this study.

### 2.4. Statistical Analysis

The number of components, type, and fluorescence intensity of CDOM for 90 soil samples were obtained by MATLAB 2020a (MathWorks Inc., Natick, MA, USA) and DOM-Fluor 1.7 toolbox analysis. Data were counted using Office (2016 enhanced version) and one-way ANOVA, significance analysis, and graphical plotting were carried out using Origin 2021 (OriginLab Inc., Northampton, MA, USA). Relationships between spectral parameters (Table 2), components, and influencing factors of DOM were analyzed using Canoco5's (Ithaca, NY, USA) redundancy analysis (RDA).

**Table 2.** Definition and significance of DOM optical indices and parameters used in the present study.

| DOM Quality Index | Definition and Significance |
|---|---|
| SVUA$_{254}$ and SUVA$_{260}$ absorbance:<br>SUVA$_{254/260}$ = a($\lambda$)/c(DOC) | a($\lambda$) is the UV–Vis absorbance at wavelength 254, 260 (mm) and r is the path-length of the optical (0.01 m), c(DOC) is the concentration of extractable DOM (mg·L$^{-1}$) [52]. |
| E2/E3: a(254)/a(365) or a(250)/a(365)<br>E2/E4: a(240)/a(420) or a(250)/a(436) | a(240, 250, 254, 365, 420, and 436) is the UV–Vis absorbance at wavelength $\lambda$(mm). E2/E3 is an indication of the degree of organic matter humification, with low values indicating low humification. E2/E4 indicates the source of organic matter, with higher values being endogenous and lower values being exogenous [53,54]. |
| Slope ratio: a($\lambda$) = a($\lambda_{400}$)exp[S($\lambda_{400} - \lambda$)] + K<br>SR = S$_{(275–295)}$/S$_{(350–400)}$ | S$_{(275–295)}$ and S$_{(350–400)}$ are the spectral slope S values in each range, respectively, and $\lambda_{400}$ is the reference wavelength [55–57]. |
| Humification index:<br>HIX = ($\sum$I$_{435–480}$)/($\sum$I$_{300–345}$) | Ex at 254 nm, the ratio of the integral values of the fluorescence intensity of Em in the range 435–480 and 300–345 nm, reflects the degree of humification of DOM. HIX < 4 belongs to biological or aquatic bacterial sources, 4 < HIX < 6 belong to weakly humified features and important recent autotrophic sources [58,59]. |
| Fluorescence index:<br>FI = Em(470/520) | Ex at 370 nm, the ratio of the fluorescence intensity of Em at 470 and 520 nm reflects the source of the DOM. Microbial activity is the main source of DOM for 1.7 < FI < 2.0, and the contribution of organisms is lower when 1.2 < FI < 1.5 [60]. |
| Biological index:<br>BIX = Em(380/430) | Ex at 310 nm, the ratio of fluorescence intensity at 380 and 520 nm for Em. BIX value reflects the ratio of albuminoid and biological components. BIX value reflects the ratio of albuminoid and biological components. Low biological fraction (0.6 < BIX < 0.7), DOM of biological or aquatic bacterial origin (BIX > 1) [59,61]. |

## 3. Results

### 3.1. DOM Overall Feature Parameters

As indicated in Table 3, the average SOC and DOC contents of fissured soil were 17.26 ± 2.7 g·kg$^{-1}$ and 16.72 ± 8.50 mg·L$^{-1}$, respectively, indicating that the average contents of HJ-1 and HJ-3 were significantly lower than those of HJ-2. The trends of specific UV absorbance (SUVA)$_{254}$ and SUVA$_{260}$ were consistent, and the aromatic and hydrophobic components of DOM in each fissure did not vary considerably, with the average maximum value at HJ-2. By contrast, E2/E3 and E2/E4 exhibited higher HJ-1 humification than the others and were more endogenous. S$_R$ characterizes molecular weight size and aromatization [55], indicating that the average molecular weight and aromatization of the three cleavages were similar, with HJ-1 being slightly higher than the others. The average FI, HIX, and BIX of the three fissures were less different, with all three fissures showing FI ≥ 1.9, 0.7 < HIX < 1, and BIX < 4, indicating that DOM was a strong autogenous source feature.

**Table 3.** Overall mean values of DOM characteristic parameters in fissured soils.

| Type | SOC (g·kg$^{-1}$) | DOC (mg·L$^{-1}$) | SUVA$_{254}$ | SUVA$_{260}$ | E2/E3 | E2/E4 | SR | FI | BIX | HIX |
|---|---|---|---|---|---|---|---|---|---|---|
| HJ-1 | 14.85 ± 6.8 | 15.34 ± 10.39 | 0.33 ± 0.08 | 0.31 ± 0.08 | 5.08 ± 1.21 | 12.9 ± 2.62 | 1.06 ± 0.22 | 1.99 ± 0.40 | 0.82 ± 0.13 | 1.59 ± 0.68 |
| HJ-2 | 19.35 ± 8.6 | 18.94 ± 6.92 | 0.37 ± 0.12 | 0.35 ± 0.11 | 4.25 ± 1.09 | 11.51 ± 1.99 | 0.90 ± 0.16 | 2.03 ± 0.36 | 0.77 ± 0.12 | 1.63 ± 0.74 |
| HJ-3 | 17.57 ± 8.2 | 16.50 ± 6.97 | 0.34 ± 0.07 | 0.32 ± 0.07 | 4.38 ± 0.60 | 12.06 ± 1.89 | 0.96 ± 0.08 | 2.03 ± 0.21 | 0.77 ± 0.15 | 1.56 ± 0.60 |
| Mean | 17.26 ± 2.7 | 16.72 ± 8.50 | 0.34 ± 0.09 | 0.33 ± 0.09 | 4.62 ± 1.07 | 12.23 ± 2.28 | 0.98 ± 0.18 | 2.02 ± 0.33 | 0.79 ± 0.14 | 1.59 ± 0.67 |

Note: The unit for SUVA$_{254}$ and SUVA$_{260}$ is L·mg C$^{-1}$·m$^{-1}$. Values are presented as mean ± SD.

### 3.2. Characteristics of DOC Content and Optical Factors of DOM in Fissured Soils

The DOC content of the upper layer fluctuated significantly, whilst that of the lower layer gradually decreased and stabilized (Figure 2b). However, the variation of SUVA$_{254}$ and SUVA$_{260}$ in HJ-1 and HJ-2 (Figure 2c,d) exhibited significant fluctuations at 35–40 cm and 50–60 cm, respectively, indicating that the aromaticity, humification, and hydrophobic components of soil in this layer were stronger than those of the surrounding (in response to the S$_R$ value of this layer in Figure 3a), and the middle and lower layers were more stable. HJ-3 demonstrated a strong response to the trend of its DOM, whilst the responses of HJ-1 and HJ-2 were poorer. The S$_R$ values of HJ-1 and HJ-2 were more different than that of HJ-3 in shallow and deep soils, whilst both tended to be relatively stable and more aromatic with the higher molecular weight of DOM in the middle layer. The variations of E2/E3 and E2/E4 (Figure 3b,c) were more consistent, but more differences were found in deep soil, indicating that the bottom DOM of HJ-1 and HJ-2 was less humified and more endogenous. HIX < 4 (Figure 4a) constitutes the vast majority, and the vast majority of the lower and middle DOM are of biogenic origin (endogenous and low humification). BIX is higher within the range of 0.7–1 (Figure 4b), indicating an extremely low authigenic source of DOM in the surface layer, and the middle and lower layers belong to the medium-intensity authigenic feature. FI ≥ 1.9 was the majority (Figure 4c), indicating that DOM was mostly of microbial origin with minimal influence from terrestrial sources (FI < 1.4).

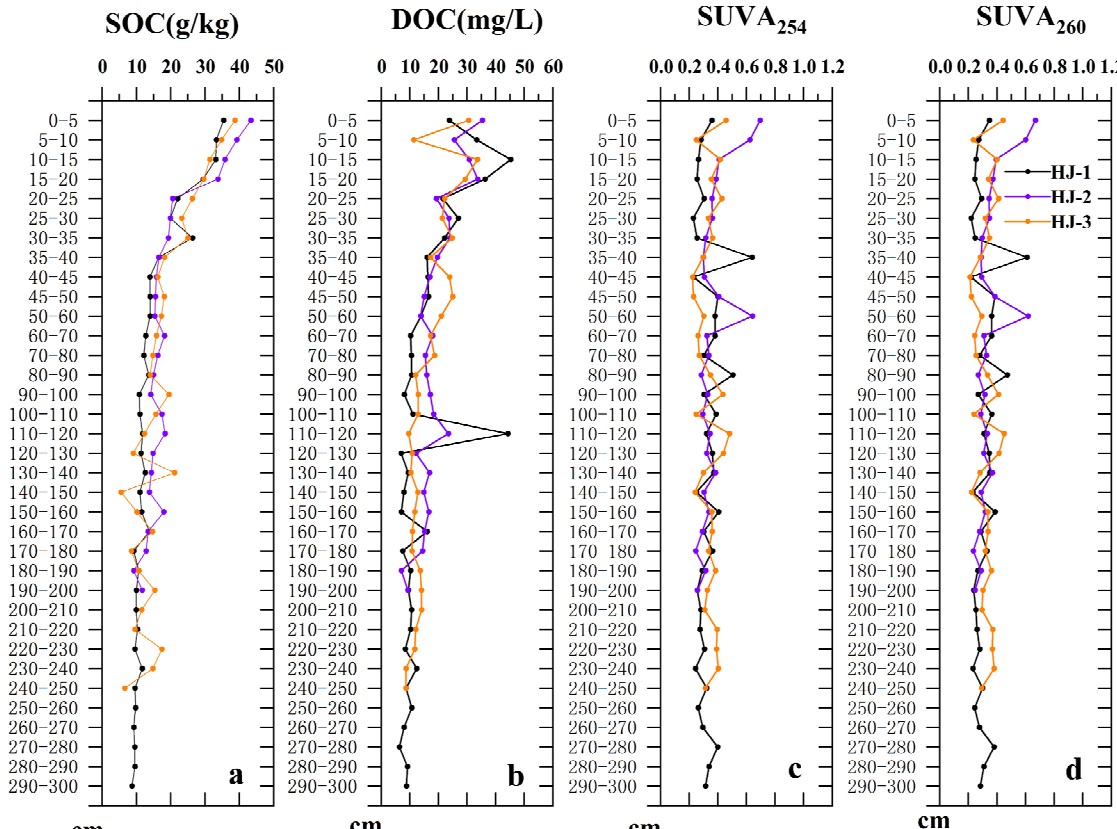

**Figure 2.** The contents of SOC (**a**), DOC (**b**), SUVA$_{254}$ (**c**), and SUVA$_{260}$ (**d**) in each layer of fissure soil. SUVA$_{254}$, SUVA$_{260}$ (L·mg C$^{-1}$·m$^{-1}$).

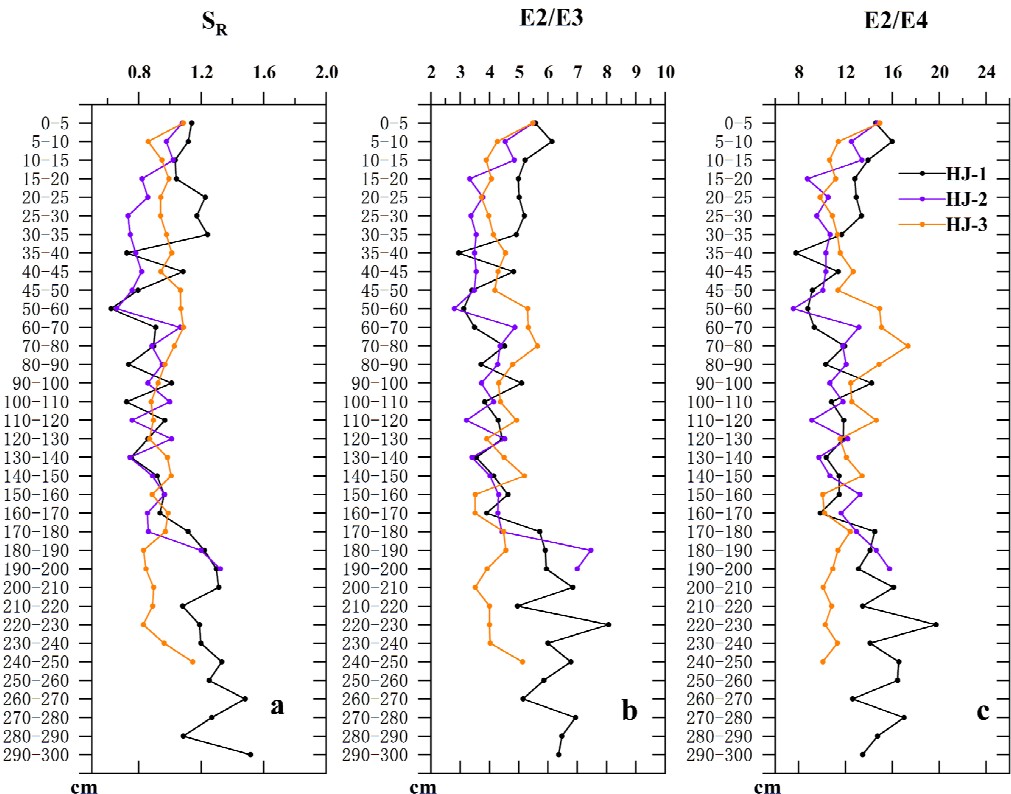

**Figure 3.** The content of E2/E3 (**b**), E3/E4 (**c**), and $S_R$ (**a**) in each layer of fissure soil.

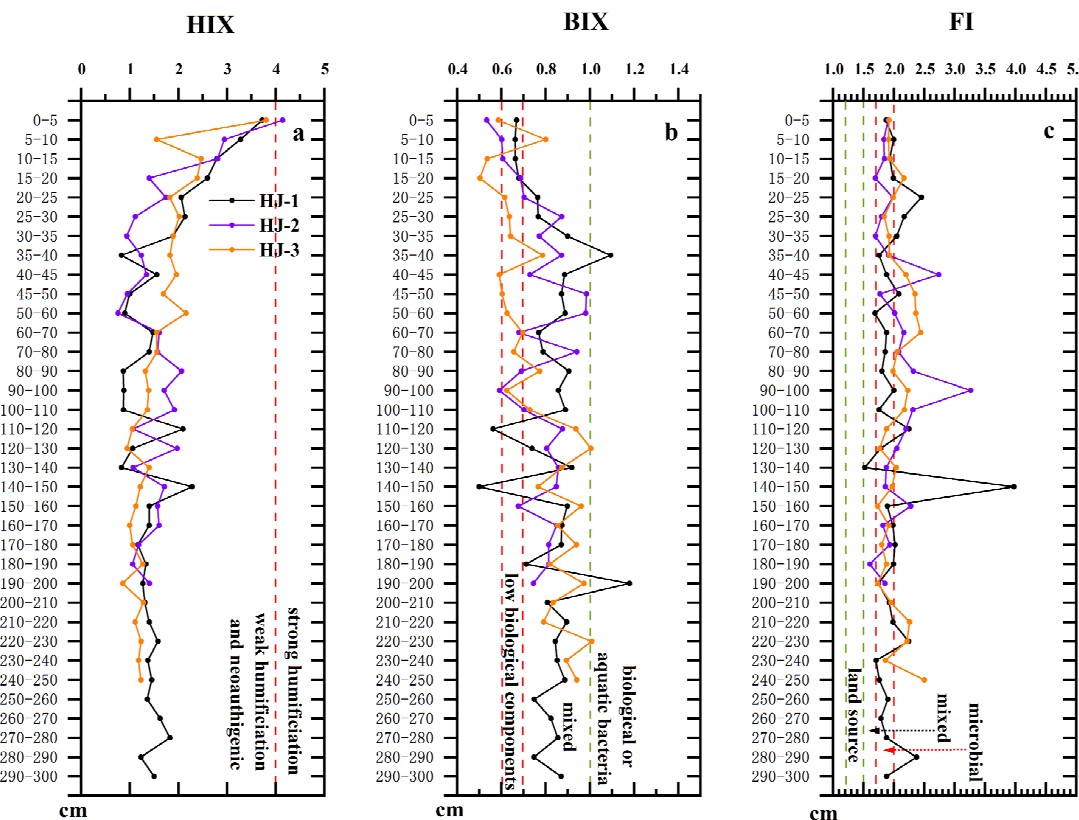

**Figure 4.** The content of HIX (**a**), BIX (**b**), and FI (**c**) in each layer of fissure soil.

### 3.3. Component Differences of DOM

Five fluorescent fractions (Figure A2) were derived for DOM in all three fissured soil samples through the EEMs–PARAFAC method. The maximum wavelengths of their fluorescence peaks were identified (Table 4), yielding C1, C2, and C5 as humic acid-like, and C3 and C4 as protein-like. Selection and comparison revealed that C1 and C5 were similar to the traditional fulvic acid A peaks (Ex: 237–260/275; Em: 400–500/<500 nm) [62,63]. C2 was identified as the traditional humic-like M peak (Ex: 290–325; Em: 370–430) [62,64]. C3 and C4 are similar to protein-like peaks (Ex: <240/275; Em: 330–368/340) [64], and C3, C4, and C5 are more evidently redshifted, implying an increase in molecular weight. As shown in Figure 5, all three clefts exhibit greater humic-like than protein-like peaks, with the largest proportion of the C1 component and the smallest proportion of the C5 component. The fluorescence intensity of all the peaks of HJ-2 is stronger than those of the others.

**Table 4.** Types of DOM fluorescent fractions in fissured soil.

| Type | Maximum Wavelength | HJ-1 | HJ-2 | HJ-3 | Description of the Source |
|------|------|------|------|------|------|
| C1 | $Ex_{max}$ | 255 | 255 | 255 | A peak, terrestrial humic [62]. High molecular weight and aromatic humus, widely distributed, highest in wetland and forest environments [65]. |
|  | $Em_{max}$ | 461 | 471 | 452 | |
| C2 | $Ex_{max}$ | 305 | 305 | 305 | M peak, marine humic [64], Low molecular weight, similar in marine, wastewater, wetlands, and farmland [63,66]. |
|  | $Em_{max}$ | 415 | 428 | 403 | |
| C3 | $Ex_{max}$ | 255 | 245 | 250 | T peak, protein-like peaks, and microbial by-product-like substances are related [64,67,68]. |
|  | $Em_{max}$ | 369 | 366 | 369 | |
| C4 | $Ex_{max}$ | 280 | 285 | 280 | B peak, protein-like Complexine [64], leachate production mainly by microorganisms, phytoplankton, and higher plants [20,63]. |
|  | $Em_{max}$ | 336 | 334 | 328 | |
| C5 | $Ex_{max}$ | 290 | 290 | 285 | A and C peak, terrestrial humus [63]. |
|  | $Em_{max}$ | 521 | 521 | 521 | |

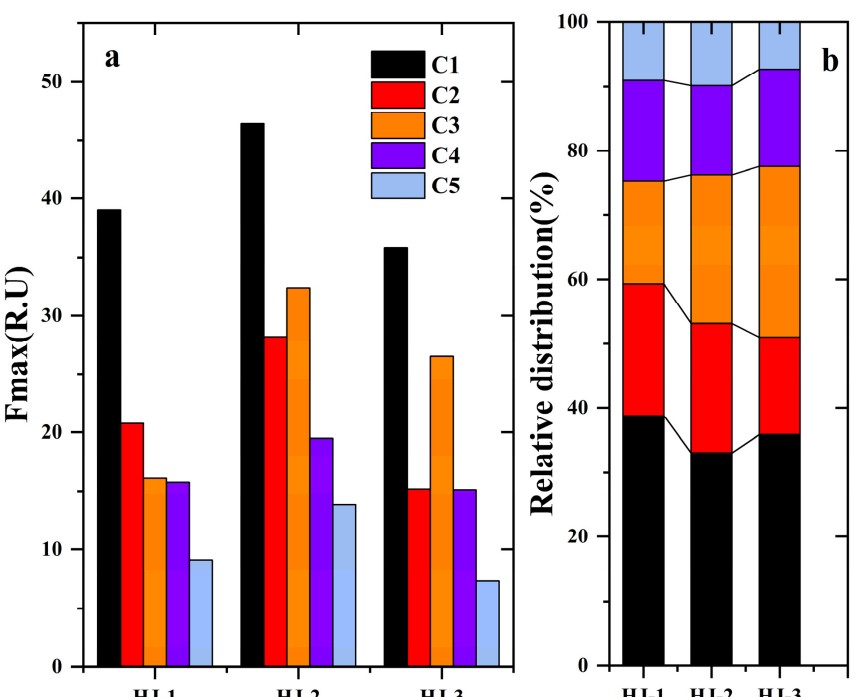

**Figure 5.** Intensity (**a**) and percentage (**b**) of fluorescence 5 components of DOM in fissured soils.

### 3.4. Factors That Affect DOM Migration

To highlight the effect of each factor on DOM migration in fissured soil, Conoco's RDA and Pearson correlation coefficients were calculated amongst parameters of interest. Environmental factors included near-surface vegetation (NSV), fracture width (FW), fracture dip angle (DA), depth (De), and aperture (Ape). Soil environmental factors were pH, volumetric weight (VW), total porosity (TPs), and electrical conductivity (EC). Soil quality factors were total P (TP), total N (TN), SOC isotope (Is), and DOC/SOC. As a soil quality standard, DOC/SOC measures the total variability of soil C pools in karst systems [69]. A conclusion could be drawn from Figure 6 that environmental factors exerted a significant effect on DOM characteristics in fissured soil, particularly De, NSV, and Ape. pH had minimal effect, whilst a significant correlation was observed between nutrient elements (TP, TN, and Is) and DOM characteristic indicators. $SUVA_{254}$ was positively correlated with the aromaticity and hydrophobicity of DOM [4,70], whilst $SUVA_{260}$ was positively correlated with the concentration of hydrophobic components of DOM [71]. $SUVA_{254}$, $SUVA_{260}$, and DOC were highly correlated with all five components (Figure 6a), allowing better characterization of DOM. De was significantly and negatively correlated with the five components (Figure 6a). NSV significantly affected the protein-like and low-humic fractions of DOM with a positive correlation ($p \leq 0.05$) and a significant negative correlation with E2/E3 ($p \leq 0.01$) and $S_R$ ($p \leq 0.05$). As shown in Figure 7, RDA in Axis 1 was 80.9% and in Axis 2 was 11.31%.

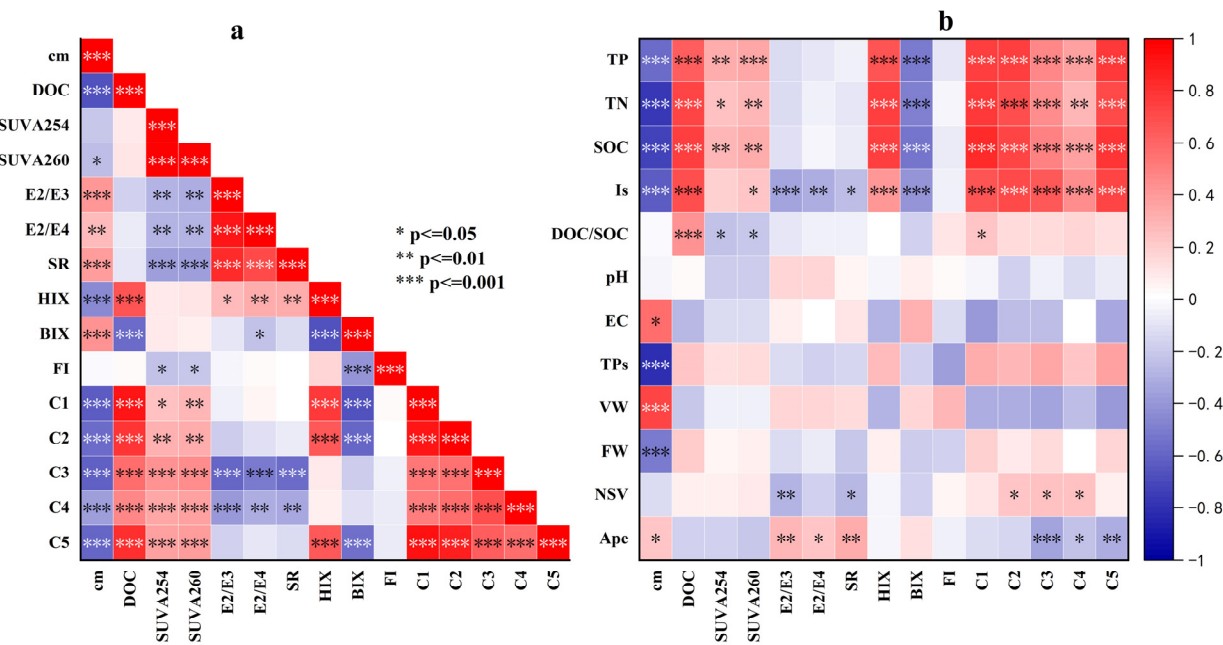

**Figure 6.** Pearson correlation coefficients between DOM parameters (**a**) and environment (**b**) for fissured soil.

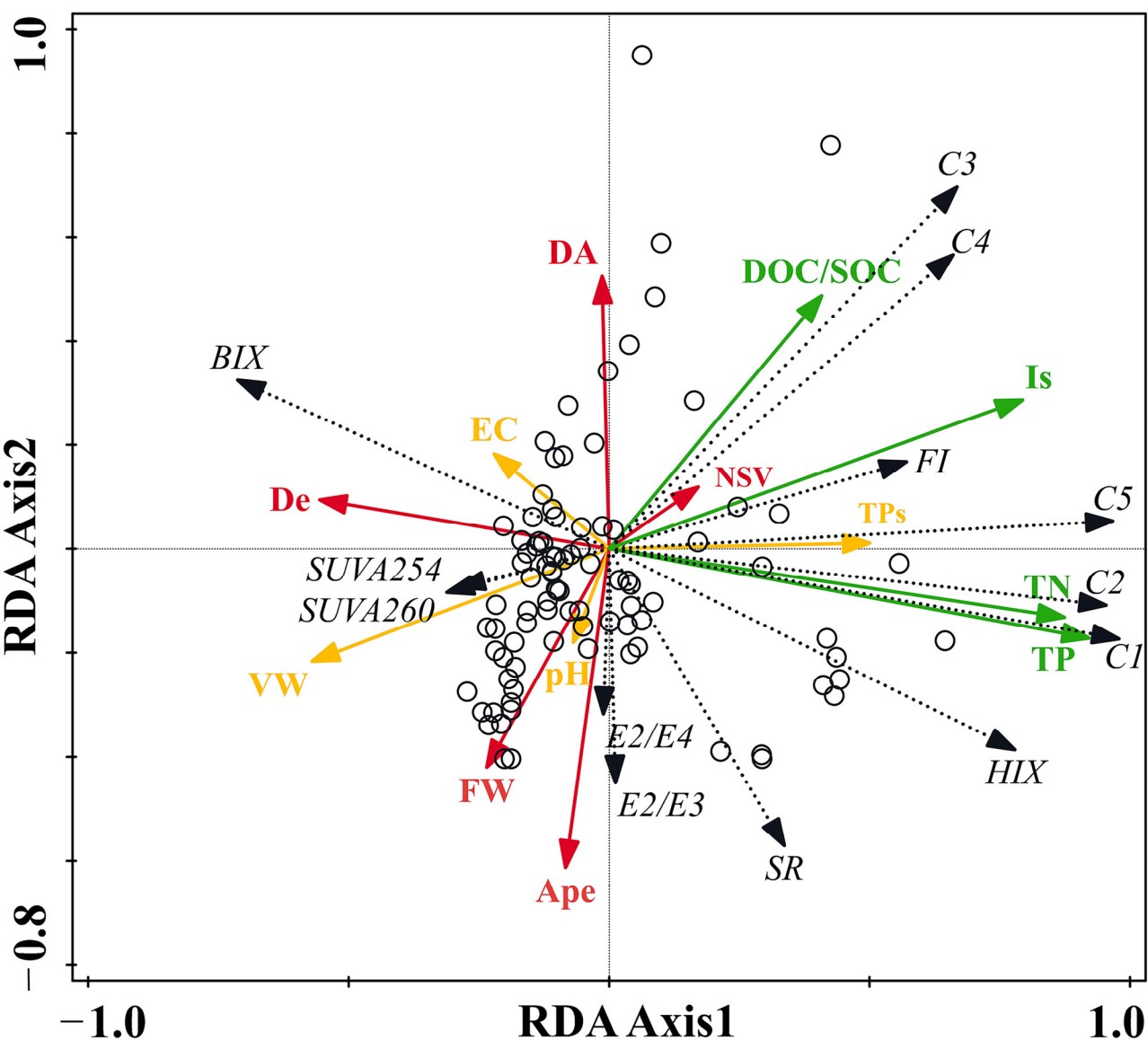

**Figure 7.** RDA ranking plots of DOM control factors (solid line) and Spectral factor properties (dashed line) in fractured soil. The red, yellow, and green solid lines represent environmental factors (NSV, FW, De, and Ape), soil physical factors (VW, TPs, EC, and pH), and soil chemistry factors (TP, TN, DOC/SOC, and Is), respectively.

## 4. Discussion

### 4.1. Characterization of SOC and DOM Content in Fissured Soil

DOM, as the most active part of SOM, is the major energy substrate of soil and it exerts an important effect on the organic C pool [72]. The shallow SOC and DOC contents of fissured soil in the study area (Figure 2a,b) were generally consistent with those of other studies of the same type [73,74], although SOC was significantly higher in the study area than in non-karst areas [47,75,76], whilst DOC was significantly lower in the study area than in non-karst areas [77]. With deeper depth, SOC and DOC contents gradually decreased and stabilized [78,79]. Therefore, a large amount of SOC storage is found in the study area, but the special binary 3D structure of the karst region has large DOM mobility and low storage capacity.

A significant correlation (Figure 6, $p < 0.001$) and a linear relationship (Figure 8) was observed between C3, C4, and C1, C2, C5, and the linear relationship between C1, C2, and C5 ($R_1^2 = 0.85$, $R_2^2 = 0.76$) was significantly stronger than that between C3, C4, and

C5 ($R_3^2 = 0.38$, $R_4^2 = 0.32$). This finding suggests a transformative relationship between the DOM components of fissured soil. Proteins in soil organic matter are easily broken down by microorganisms into monomers, which, in turn, collect in the soil to form humus through microbial activity [80]. Therefore, C3 and C4 may be converted into C2 (medium molecular weight, Table 4) and then into C1 and C5 (larger molecular weight, more stable), because soil microorganisms preferentially utilize unstable components (e.g., tryptophan and tyrosine); when the content of unstable components (C3 and C4) is low, microorganisms may utilize the more stable C1, C2, and C5 [81]. The average protein content of fissured soil in the study area was 36.78% ± 0.05% (Figure 6), which was 115–143% higher than that of the surface layer of agricultural soil in Gao [77], including Heilongjiang, Jilin, Liaoning, Shaanxi, Shanxi, Hebei, Tianjin, Henan, Shandong, Chongqing, Hunan, Anhui, Jiangsu, Jiangxi, Gansu, Xinjiang, Inner Mongolia, and Yunnan Province. This finding indicates that amino acids or degraded peptides are more abundant in the fissured soil DOM in the study area, which may be attributed to humus in a subtropical environment contributing to the enrichment of protein substances [82]. Thus, the decomposition rate of protein-like substances or the content of incompletely degraded peptides was relatively high in the fissured soil of the study area [77].

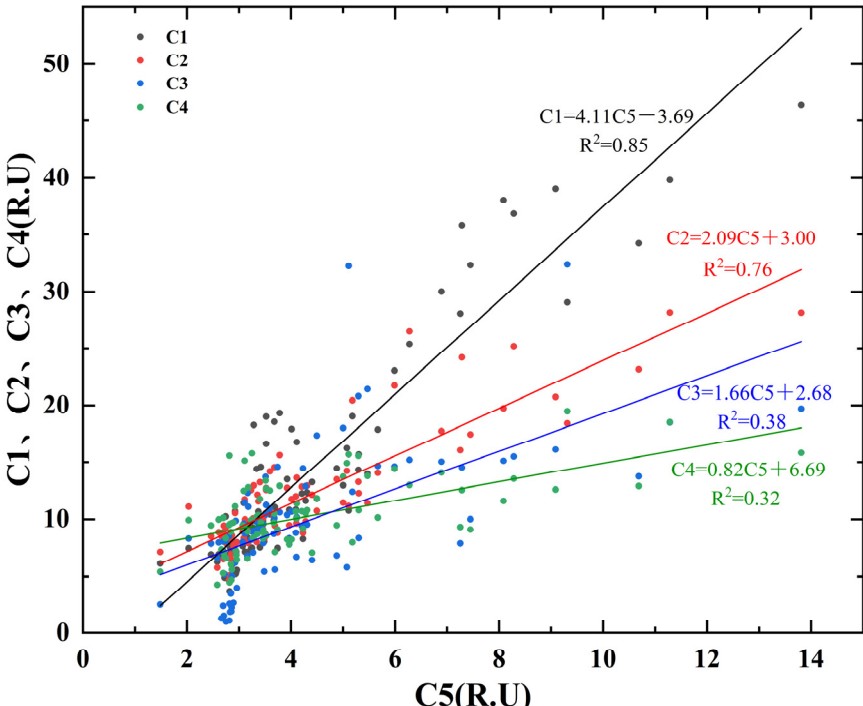

**Figure 8.** Relation of C5 to C1, C2, C3, and C4.

As shown in Figures 4 and A1, the surface soil exhibits a low authigenic phenomenon via external influence, whilst the middle and lower soil DOM are overwhelming from authigenic or microbial sources with a low degree of humification [58]. The higher content of DOM-like proteins in fissured soil is due to the higher material circulation capacity and reduced conversion of protein-like substances into humus during the rainy season. The low DOM content of karst areas reduces microbial activity [23], slows down the effective utilization of protein-like substances, and weakens the mineralization and utilization of low-molecular-weight organic matter [47] by the soil, making losing these nutrients through fissures easier.

*4.2. Drivers That Influence Changes in DOM Components in Fissured Soil*

In the study area, the DOM of fissured soils is likely to be strongly influenced by environmental and vegetation factors. The overview of the study area noted that its

average annual rainfall is 1052 mm, which will seriously affect the distribution of DOM [83]. Rainfall intensity affects the migration of DOM in fissured soils. A previous study [34] showed that DOC distribution in fissured soil in the study area was significantly influenced by hydrological transport: (1) The performance of $\delta$D and $\delta^{18}$O enrichment in the study area indicates that rainfall to soil water undergoes the transport process of rainfall $\rightarrow$ penetrating rainfall $\rightarrow$ rock wall flow $\rightarrow$ apoplastic material $\rightarrow$ soil water when rainfall intensity is moderate (24 h rainfall: 10.0–24.9 mm). In excess of 35 mm rainfall (heavy rainfall), only the lower and middle layers of fissured soil will respond; and (2) Changes occur in DOC before the rainy season (January–May) and after the rainy season (June–September) with an average reduction rate of 21.14%. As important water-conducting channels in karst areas, fissures provide the link between atmospheric precipitation, surface water, soil water, and surface and underground rivers, playing an important role in water transport in karst areas [84]. It can lead to aggravated DOM loss.

Through previous studies [49] and field surveys [34,85,86], the study area is heavily karst desertification and is of medium intensity (up to 70% rock exposure). This is not conducive to DOM accumulation. The higher the rate of rock exposure, the more easily soil is washed away; this condition is not conducive to the accumulation of aromatic and hydrophobic substances in soil [86]. Although exposed rocks will enrich the water and nutrients of nearby soil, excessive water input during heavy rainfall leads to severe erosion and soil seepage, reducing the bearing capacity of soil [87]. As the degree of fissure opening, the Ape directly influences the infiltration and transport of soil water and nutrients, affecting the accumulation rate of the soil, gravel, and other fillers. This explains the greater loss of the lighter components of its DOM as the Ape increases (Figure 6).

In accordance with Figure 6, NSV significantly affected the C2, C3, and C4 fractions of DOM. This result related to the physiological and biochemical effects of vegetation roots and soil microbial activity, because the intervention of vegetation significantly promotes fast-acting nutrients in the cleaved area, making the fast-acting nutrient content of the soil in the cleaved area significantly higher than that in the non-cleaved area [88]. Consequently, the percentage of protein-like substances in the DOM of cleaved soil in the study area was higher. Vegetation within fissures changed from herbaceous to tree; hence, differences in their soil nutrient status were found. Organic matter tended to increase and gradually decreased significantly with an increase in soil depth. The organic matter and TN content of the soil surface layer were significantly higher than those of the lower layer [89]. The differences in DOM characteristics [90,91] were caused by differences in organic matter input and root secretions from different plant sources. The reasons for the differences in the DOM fraction content of cleaved soil caused by different overlying plants were explained.

### 4.3. Indicative Significance of DOM Distribution in Fractured Soil

The migration of the DOM has caused a change in the environment. A correlation exists between DOC and various nutrients, such as N, P, and K in soil (Figure 6), particularly P. DOC increases the mobility of P to occur as Ca–Mg–P rather than as the more insoluble hydroxyapatite [92]. DOC can increase the use of P by plants (crops), and the increased mobility of P makes leaching easier [88]. The higher $Ca^{2+}$ and $Mg^{2+}$ contents in karst areas enhance the mobility of P more easily, possibly increasing damage to surface water and groundwater.

Soil properties can alter the extent of DOM loss. Numerous studies [93,94] have shown that conductivity can determine the concentration of nutrients, salt-based ions, and other solutes in soil, along with the content and migration processes of solutes. Soil EC in the study area was significantly higher during the rainy season than during the non-rainy season [34]. VW was significantly higher in deeper soil than in the surface layer, and soil porosity decreased with depth (Figure 6), which might be the result of rainfall migrating water-carrying nutrients and other salt-based ions from fissured soil. This finding is also in line with the fact that low bulk weight, high porosity, and sand and powder content improve the permeability of the soil and facilitate soil water transport [34]. If fissure flow is

present in the soil, it can increase the loss of soluble nutrients in the soil [37]. In addition, the irregularity of rocks in karst areas causes differentiation in the infiltration of rainfall at different rock–soil interfaces [95], resulting in inconsistent nutrient migration and degree of loss in different fissures; this finding is related to soil properties and water transport effects.

In summary, DOM plays an important role in the soil–water system as a major source of C and nutrients [96,97]. The risk of loss of fissured soil DOM is greater in the study area. This finding is largely related to rainfall intensity, near-surface plant type, degree of fissure development, and soil condition. Research on the effects of vegetation types and soil microorganisms on DOM should be deepened, and more DOM characteristics in fissured soils of karst areas should be analyzed to improve the karst desertification environment. The transport distribution of DOM not only exerts an important effect on the C cycle but also on the transport of nutrients. The biogeochemical cycling and transport of DOM with heavy metals and persistent organic pollutants cannot be disregarded [98–102]. Considering that fissures are amongst the leakage channels and storage spaces of soil–water–nutrients in karst areas, DOM should be the focus of research.

## 5. Conclusions

The source, distribution, and influencing factors of DOM in karstic fissured soils were obtained in this study, leading to three conclusions: (1) The DOM of the fissured soils in the study area, with the exception of the surface layer, which is subject to higher values of external environmental influences, has a lower DOM content of the middle and lower layers, and is less aromatic and less humified than the surface layer; (2) Rainfall intensity in karst areas severely controls the distribution and loss of DOM in rift soil. Under the synergistic influences of higher rock exposure rates and NSV, more protein-like fractions may be preferentially lost. In addition, fissure morphology and soil properties affect DOM loss to some extent; and (3) DOM loss in fissured soil not only affects soil nutrient loss but also increases soil P mobility, particularly in karst areas.

Therefore, areas with high karst desertification intensity, strong fissure development, weak aromaticity, and low humification of DOM in fissured soil, along with the preferential loss of low-molecular-weight components of DOM and the downward migration of other nutrients and solutes, severely affect soil quality and water ecosystems, providing a basis for understanding the optical properties of DOM in karstic fissured soil and influencing factors.

**Author Contributions:** Conceptualization, K.X. and K.C.; methodology, K.C.; software, formal analysis, and data curation, K.C.; resources, K.X., Z.L. and Q.H.; investigation, L.C., Y.C., Y.L. and K.C.; writing—original draft preparation, K.C.; visualization, K.X. and K.C.; writing—review and editing: K.C., Z.L., K.X., Q.H., Y.L., L.C. and Y.C.; supervision, K.X. and K.C.; project administration and resources, Z.L. and K.X.; funding acquisition, K.X. and Z.L. All authors have read and agreed to the published version of the manuscript.

**Funding:** This work was supported by the Philosophy and Social Science Planning Key Project of Guizhou Province (Grant No. 21GZZB43), the Key Project of Science and Technology Program of Guizhou Province (Grant No. 5411 2017 Qiankehe Pingtai Rencai), and the China Overseas Exper-tise Introduction Program for Discipline Innovation (Grant No. D17016).

**Data Availability Statement:** Not applicable.

**Conflicts of Interest:** The authors declare no conflict of interest.

**Appendix A**

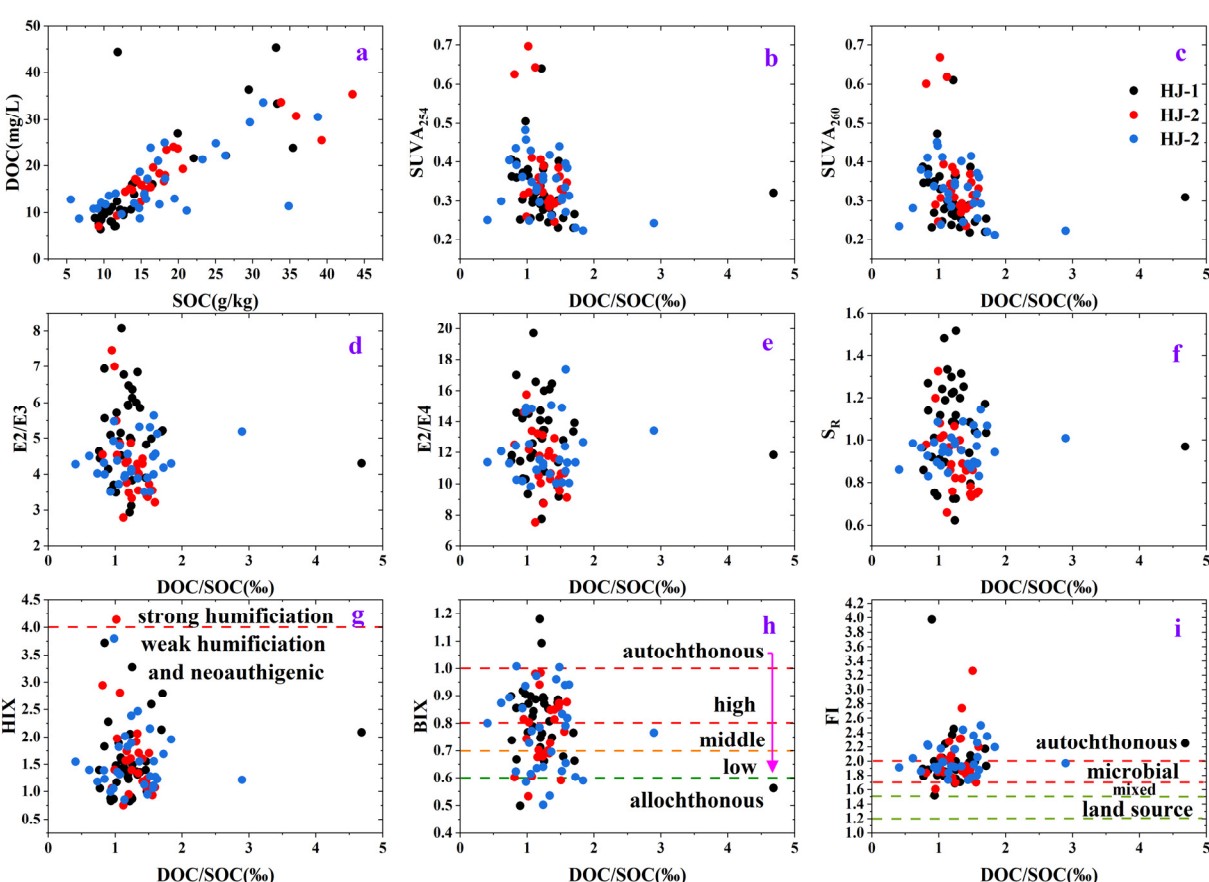

**Figure A1.** Scatter plots of SUVA$_{254}$ (**b**), SUVA$_{260}$ (**c**), E2/E3 (**d**), E2/E4 (**e**), FI (**i**), S$_R$ (**f**), HIX (**g**), and BIX (**h**) with DOC/SOC (‰) as the *x*-axis, and DOC (**a**) along SOC. SUVA$_{254}$ and SUVA$_{260}$ in L·mg C$^{-1}$·m$^{-1}$.

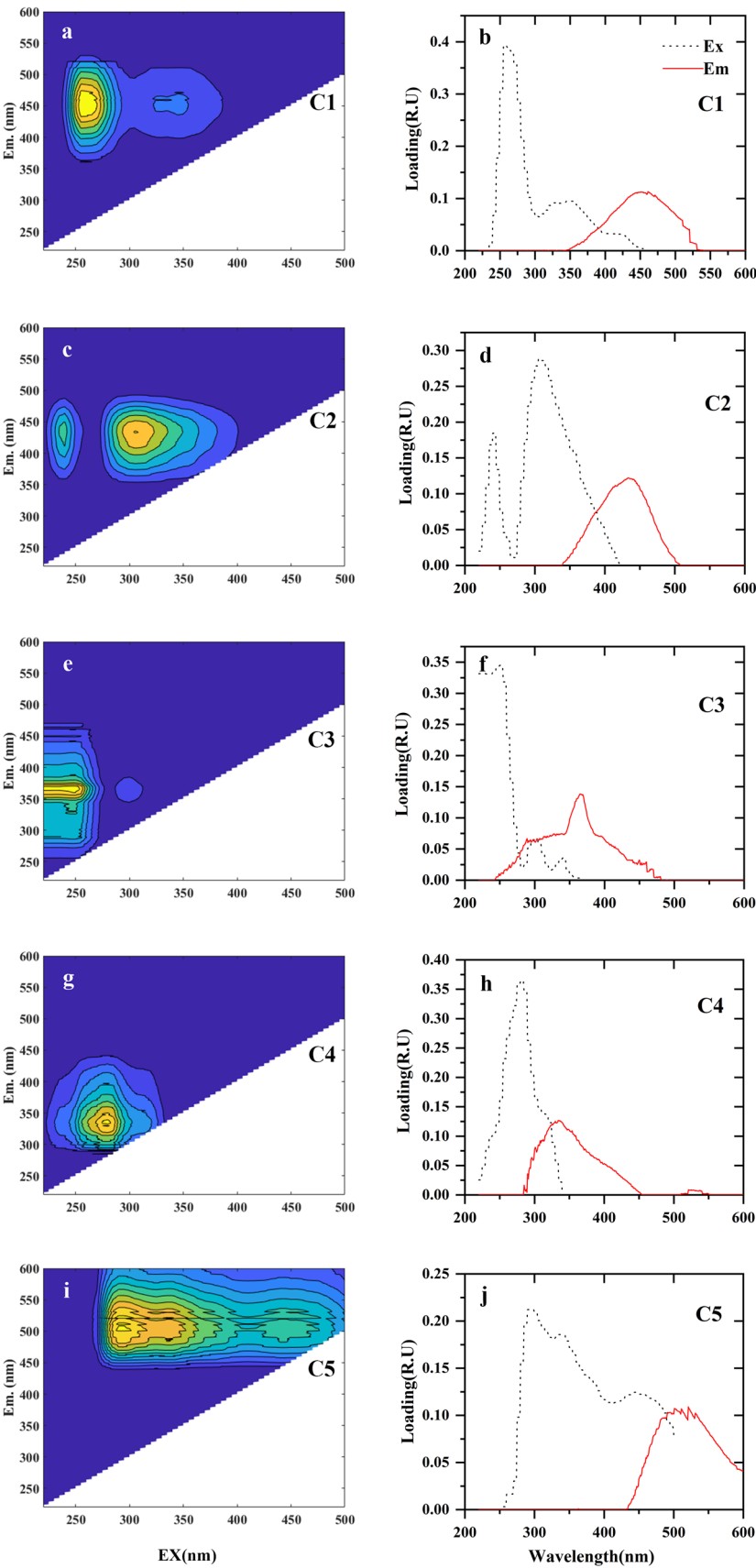

**Figure A2.** Contour plots of the five fluorescent components of DOM in soil samples derived using EEMs-PARAFAC, where C1 (**a**,**b**), C2 (**c**,**d**), C3 (**e**,**f**), C4 (**g**,**h**), C5 (**i**,**j**).

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
