# Peer review of "Migration of Dissolved Organic Matter in the Epikarst Fissured Soil of South China Karst"

_land, doi:10.3390/land12040887_

Round 1
Reviewer 1 Report
This paper aim to characterize DOM in the fissured soil of different vegetation types under medium-intensity rocky desertification condition. Soil samples were characterised via ultraviolet (UV)–visible absorption spectroscopy and fluorescence excitation emission matrix–parallel factor analysis. Five fluorescent fractions were identified. The controlling factors for the optical properties of soil DOM were determined via the redundancy analysis method. I think the paper provide a clearer basis for optimising the fissure nutrient element migration scheme in karst areas. I think the paper can be accepted with major revisions. There are some suggestions as follows:
1. The DOC and DOM was confused in the whole paper. The DOC not equal to DOM.
2. Line 134 why the 0.22μm polyethersulfone membrane was used? Not 0.45μm?
3. Line 183, the DOM content should be DOC content
4. Line 226, affect changed to affecting; migration changed to migration
5. Line 256, DOM should be a active part of SOM.
6. Line265-267, delete the sentence “Through……..(C3 and C4)”, this is a result description.
7. Line287-292, the sentences are results and should not be here.
8. Line376-378, the paragraph is confusing me.
Reviewer 2 Report
This study adds to our understanding of the content and spectral characteristics of DOM typical fractured soils from various vegetation types under desertification conditions. Furthermore, the study's findings aid in understanding the transport distribution of DOM in rift soils of Karst areas, as well as its relationship to soil carbon and nutrient dynamics. The manuscript is well written, and the scientific results of related works are presented and explained adequately. Figures and tables are simple to understand. The introduction is based on recent discoveries, and the "Materials and Methods" section is thoroughly described. In general, the main conclusions and ideas are adequate and justified. In the following, a few minor changes are suggested (please also see the attached file).
Line 180: Table 3 Please format the size of the table.
Line 348: Change the format of….. higher Ca2+ and Mg2+ contents....
Lines 365-375: Please move the paragraph to Section Conclusions.
References must be formatted in accordance with the journal's guidelines.
The manuscript is recommended for publication in this journal, considering minor changes.

Reviewer 3 Report
Dear authors,
The manuscript is interesting and should be published with minor changes. The main objective is the reactivity and mobility of dissolved organic matter in fissured soil.
I suggest to add a short conclusion in order to be clearer the results, innovation and suggestion of their study.
At the paragraph 2.2 explain in few words how was collected the soil samples for the organic matter in order to be representative of the area.
Line 226 correct “migration”
Line 266 delete “three” you write it two times.
So, I suggest that it should be published with minor changes
Sincerely yours,
Round 2
Reviewer 1 Report
I think the paper should be accepted.